# Mixture of Tomato and Lemon Extracts Synergistically Prevents PC12 Cell Death from Oxidative Stress and Improves Hippocampal Neurogenesis in Aged Mice

**DOI:** 10.3390/foods11213418

**Published:** 2022-10-28

**Authors:** Ji Yeon Hong, Jae-Jun Ban, Qing-Ling Quan, Ji-Eun Eom, Hee Soon Shin, Jin Ho Chung

**Affiliations:** 1Department of Dermatology, Chungnam National University Sejong Hospital, Sejong 30099, Korea; 2Institute of Human-Environment Interface Biology, Medical Research Center, Seoul National University, Seoul 03082, Korea; 3Department of Dermatology, Seoul National University College of Medicine, Seoul 03080, Korea; 4Food Functional Evaluation Support Team, Korea Food Research Institute, Wanju 55365, Korea; 5Food Biotechnology Program, Korea University of Science and Technology, Daejeon 34141, Korea; 6Division of Functional Food Research, Korea Food Research Institute, Wanju 55365, Korea; 7Institute on Aging, Seoul National University, Seoul 03082, Korea

**Keywords:** aging, cognition, hippocampus, neurogenesis, nutrition, tomato, lemon

## Abstract

Dietary habits have a great impact on one’s health, especially in cognitive decline. Tomato and lemon contain diverse bioactive compounds and possess various effects, including the enhancement of cognitive function. We observed the protective effect of tomato, lemon extract and the mixture of them on H_2_O_2_-induced cytotoxicity of PC12 cells. To measure the in vivo effect in a murine model, each extract was orally administered to forty 1-year-old mice for 6 weeks, and a novel object recognition (NOR) test was performed to observe cognitive function, and hippocampal neurogenesis was observed through a doublecortin (DCX) stain. PC12 cell death by oxidative stress was reduced by pretreating with each extract, and a synergistic reduction was observed in the mixture. Newly generated DCX-positive neurons were synergistically increased in the hippocampus by the mixture. NOR test showed a tendency to significantly improve age-related cognitive dysfunction by consuming the mixture of tomato and lemon. In conclusion, tomato and lemon extracts can reduce cellular oxidative stress and increase NOR, likely due to enhanced neurogenesis, while the mixture of the two showed synergistic anti-oxidative effects and hippocampal neurogenesis.

## 1. Introduction

Cognitive decline is a hallmark of the normal aging process [1]. With the aging of the world population, dementia and cognitive decline have become enormous challenges to the sustainable development of the economy and society. Geriatric changes in cognitive function are considered to be associated with reduced hippocampal neurogenesis, which plays a major role in memory formation and can be damaged by cumulative oxidative stress [2,3]. To reverse this phenomenon, various kinds of therapeutics are available in the market with diverse opinions concerning the actual effect [4,5,6].

There is a proverbial saying that you are what you eat, which highlights the close relationship between dietary habits and health. Typically, a high-calorie diet with much fat and salt can cause cerebrovascular diseases and metabolic syndrome [7]. Conversely, Mediterranean diets, including plant-based foods, are known to reduce the incidence of Alzheimer’s disease and the risk of cognitive decline [8]. It has been suggested that the improvement in cognitive function related to aging may be due to the antioxidants and vitamins present in the diet [9].

Tomato (*Lycopersicon esculentum* Mill) is a fruit food that is widely consumed worldwide and contains many bioactive compounds such as vitamin C, polyphenols, flavonoids, anthocyanins, and carotenoids [10]. It has been reported that lycopene contained in tomato alleviates cognitive decline induced by diabetes [11], high-fat diet, and Parkinson’s disease in rodent models [9,10]. In addition, long-term intake of tomato extract can enhance hippocampal neurogenesis and cognitive function and augment the brain-derived neurotrophic factor (BDNF) signaling pathway in aged mice [12]. Lemon (*Citrus limon* (L.) Burn f.) is the most consumed citrus species, followed by orange and mandarin. Lemon contains phenolic compounds, vitamins, minerals, and carotenoids, and the compounds contained in lemon have antioxidant and anti-inflammatory properties and neuroprotective effects [12,13]. It has also been reported that hesperidin contained in *Citrus limon* can attenuate cognitive impairment by reducing oxidative stress, mitochondrial dysfunction, and apoptosis [14,15].

Although independent positive effects of tomato and lemon on cognitive function have been reported, the combined effects of long-term consumption of a mixture of these two fruit ingredients have not been studied yet. Tomato and lemon both are quite easily accessible and commonly consumed food over the world. However, little is known about what the merits or demerits are to have them together especially in dealing with cognitive function. To investigate the synergistic effects of orally administered tomato and lemon extract in aging-associated cognitive decline, we analyzed hippocampal neurogenesis and cognitive function of 1-year-old mice after feeding the extracts for 6 weeks. 

## 2. Materials and Methods

### 2.1. Preparation of Tomato and Lemon Extract

All methods for preparing plant extracts in this study complies with the guidelines of Korean Food and Drug Administration. For washing and sterilization of tomatoes and lemons, the raw materials were washed by mixing with distilled water approximately 20 times the weight. Distilled water was removed, and the raw material was recovered and subjected to hot air drying. The dried tomato and lemon were mixed with 70% alcohol 10 times by weight, and alcohol extraction was performed with intermittent stirring for 6 h at 80 °C for tomato, and 80 °C for 4 h (1st extraction) and 2 h (2nd extraction) for lemon. After concentration, the same amount of dextrin was added and sterilized by heating (95 °C for 1 h). Finally, spray drying was performed under the conditions of inlet air temperature of 180 °C and outlet temperature of 85 °C to obtain the powdered extract.

### 2.2. PC12 Cell Culture

The PC12 cell line, derived from a rat pheochromocytoma, was obtained from the Korean Cell Line Bank (Seoul, Korea) and cultured in RPMI 1640 medium (Welgene, Gyeongsan, Korea) supplemented with 10% horse serum (Gibco, New Zealand), 5% fetal bovine serum (Welgene), and antibiotics (100 units/mL penicillin, 100 μg/mL streptomycin; Welgene). The medium was replaced every 2 days, and the subculture was repeated for four days. The cells were incubated in a 5% CO_2_ humidified incubator at 37 °C and were seeded at 1 × 10^6^ cells/well in 96-well culture plate (pre-coated with collagen) for 24 h.

### 2.3. Cell Viability Assay

Cell viability was assessed using a colorimetric method based on WST-1 (Cellvia, Abfrontier, Seoul, Korea) according to the manufacturer’s protocol. Experiment was repeated three times and three replicates were used within a plate to confirm the reproducibility. Briefly, PC12 cells (1 × 10^6^ cells/well) were dispensed into 96-well culture plates and incubated for 24 h. The cells were pre-treated with each sample extract (T, tomato; L, lemon; Mix, tomato and lemon mixture) for 24 h and then incubated with 100 μmol/L of hydrogen peroxide (H_2_O_2_) for 30 min. Then, 0.05% of tomato extract, 0.05% of lemon extract, 0.05% of mixed extracts, 0.1% of tomato extract, 0.1% of lemon extract and 0.1% of mixed extracts were used for the experiment. After incubation, 10 μL of WST reagent was added to each well of the plate, and the mixture was incubated for 3 h at 37 °C. Absorbance was measured at 450 nm using a microplate reader (Epoch, BioTek, Winooski, VT, USA). Cell viability was calculated as a percentage using the following formula:Viability (%) = {(sample-treated well OD − blank OD)/(normal well OD − blank OD)} × 100

### 2.4. Oral Administration of Tomato, Lemon, and Mixture Extract to Mice

Twelve-month-old and 8-week-old female albino hairless mice (Skh-1, average body weight 44.44 ± 3.67 g) were purchased from Orient Bio (Seongnam, Korea). We chose female mice to reduce inter-individual difference and to avoid damage or loss of mice due to male’s aggression. Hairless mice were also convenient to check physical aging by observing skin change. Animals were fed ad libitum, and all experimental protocols were approved by the Institutional Animal Care and Use Committee (IACUC) of the Seoul National University Hospital. As approved in IACUC protocol, euthanasia was performed with overdose of isoflurane and confirmed by bilateral thoracotomy. All methods were carried out in accordance with the committee’s guidelines and regulations. All the authors complied with regulations and guidelines relating to the use of animals for scientific purposes, adhering to ARRIVE guidelines. The 1-year-old mice were randomly divided into four equal groups for the vehicle control and extract-administered groups. Each group was composed of 8 mice. Eight 8-week-old female hairless mice were used as control. The animals were fed orally at a dose of 400 mg/kg once daily for 6 weeks using a feeding needle (400 mg/kg of tomato; 400 mg/kg of lemon; 200 mg/kg of tomato plus 200 mg/kg of lemon). For vehicle-treated young and aged mice, an equal volume (200 µL) of 0.5% carboxymethyl cellulose–sodium solution was administered once daily for 6 weeks. We selected the dose of 400 mg/kg considering the potential development of the extracts as dietary supplement for human if the study results showed remarkable improvement. The dosage of extracts was decided considering the amount of human intake and the compliance. The dose of “400 mg/kg” in mouse can be translated to 2 g/day in 60 kg weighted human [13]. Generally, the tablet of one-fingertip size weighs 500 mg. That is, 2 g per day equal to 2 tablets twice daily, which shows clinically suitable compliance.

### 2.5. Novel Object Recognition Test

A novel object recognition (NOR) test is a widely used model for investigating memory alterations [14]. In NOR test, there are no rewards for the animal and it explores the novel object just as its natural propensity and innate exploratory behavior to the novelty. In our study, the test was performed using a modified method with plastic apparatus, according to the verified method of Clarke, et al. [15] We conducted pilot experiments to confirm that animals do not show a preference for either objects. Briefly, the test setup consisted of an opaque rectangular plastic apparatus and the procedure consisted of three stages: habituation, training, and testing. The mice were allowed to navigate the chamber freely without objects for 5 min to get used to the environment (habituation). One day later, the mice were exposed for 5 min using the same pair of objects (training). To measure short-term memory after 1 h of training, one object was replaced with a novel object, and the mouse was placed in the chamber and recorded for 5 min (test). The time at which the mouse showed interest in the familiar object or new object was measured. The testing sessions were recorded using a video camera and analyzed by a blinded examiner. Object recognition was defined as the case in which the mouse’s nose touched the object or entered within 2 cm of the object.

### 2.6. Immunohistochemistry

After the novel object recognition test, mouse brains were isolated, and immunostaining for doublecortin (DCX) was performed using a free-floating technique. Mouse brains were removed, post-fixed using 4% paraformaldehyde overnight, and immersed in a 30% sucrose solution. Serial 30 µm coronal sections of the hippocampus were cut on a freezing microtome (Leica) and stored in cryoprotectant (35% ethylene glycol, 25% glycerol, 0.05 M PBS). Three brain sections per mouse were randomly selected for staining. To avoid selection bias and minimize topographical or dimensional differences between the pieces, we chose 1 per 10 section pieces. Three sections were analyzed per mouse—eight mice were dissected and total of 24 sections were examined for DCX cells. 

The pre-floating slices were blocked with a protein block solution (GBI Labs). Subsequently, sections were stained with an antibody against DCX (sc-8066, Santa Cruz Biotechnology, Dallas, TX, USA) overnight at 4 °C. Sections were incubated with rabbit anti-goat IgG secondary antibody (Vector Laboratories) for 2 h at room temperature. The antibody against DCX was diluted to 1:200, while the secondary antibody was diluted to 1:300. Sections were then incubated with Vector ABC kit (Vector Laboratories). DCX-positive cells were visualized with 3,3'-diaminobenzidine (Vector Laboratories) and images were captured using a Leica DM5500B microscope (Leica). To quantify the total number of DCX-positive cells in the hippocampal dentate gyrus, cell counts were performed in at least three consecutive sections at a magnification of 40× using a counting grid, defining an area of interest to a width of 500 μm along the fissure by a blinded examiner.

### 2.7. Statistical Analysis

Cell viability data were analyzed for significant differences using one-way ANOVA, followed by Tukey’s multiple comparison test. Immunohistochemistry data were analyzed using one-way ANOVA with Dunnett T3 post-test for multiple comparison trials. Behavioral data were analyzed for significant differences using one-way ANOVA, followed by Fisher’s least significant difference test as a multiple comparison test. *p*-values less than 0.05 were considered statistically significant. Results are expressed as mean ± standard error of the mean.

## 3. Results

### 3.1. Synergistic Cytoprotective Effect of the Mixture of Tomato and Lemon Extracts against Oxidative Damage in PC12 Cells

To identify the cytotoxic dose of tomato or lemon extract or a mixture of them on PC12 cells, we measured cell viability after treatment with each extract using the WST-1 assay. We observed that 0.1% of each extract did not affect cell viability, although treatment with the extract at a dosage above 0.5% started to decrease cell viability (data not shown). To assess the cytoprotective effects of the extracts on PC12 cells against oxidative stress, the cells were pre-treated with tomato, lemon extract, or the mixture for 24 h and then treated with 100 μmol/L of H_2_O_2_ for 30 min. Cell viability was determined using WST-1 assay.

Treatment with H_2_O_2_ significantly reduced the cell viability by an average of 49.0 ± 7.4%, compared with the untreated control cells (*p* = 0.047, Figure 1). Surprisingly, while 0.05% of each extract did not exhibit a protective effect (relative cell viability was 50.7 ± 6.8% and 53 ± 15.1% for tomato and lemon extract, respectively), pretreatment with the mixture (0.05%) significantly prevented H_2_O_2_-induced cytotoxicity, recovering cell viability to 80.1 ± 24.0%. Pretreatment with 0.1% tomato or 0.1% lemon extract significantly prevented H_2_O_2_-induced cytotoxicity, showing increased relative cell viability of 78.5 ± 14.2% and 80.4 ± 15.9%, respectively (*p* = 0.533 and *p* = 0.457 compared with H_2_O_2_-‘treated cells). Moreover, by preconditioning with a mixture of 0.1% extracts, PC12 cells were recovered from H_2_O_2_-induced cellular toxicity (106.0 ± 15.9%, *p* = 0.021 compared with H_2_O_2_-treated cells) nearly to the viability level of the untreated normal cells. The *p*-value of ANOVA test was 0.004 (between groups) with df = 7 and F(7, 16) = 4.819. 

### 3.2. Enhancement of Cognitive Function in 1-Year-Old Mice by Tomato, Lemon, and Mixture of Both Extracts

A novel object recognition test was performed after oral administration of tomato or lemon extract or the mixture of both extracts for 6 weeks. The longevity to show curiosity toward new objects reflects the cognitive function of the mouse. The duration of exploring novel objects for old mice (4.27 ± 1.38 s) was significantly reduced compared with that of young mice (10.09 ± 3.68 s, *p* < 0.001, Figure 2). The tomato extract-treated group (4.37 ± 2.16 s, *p* = 0.94) and lemon-treated group (4.68 ± 2.11 s, *p* = 0.73) did not show statistically significant increase in the time of interest for the new object, compared to that of the old control group. Similar to the result of hippocampal neurogenesis analysis, the mixture of tomato and lemon extracts showed significantly increased duration of interest in the new objects (6.81 ± 2.06 s, *p* = 0.04). The *p*-value of ANOVA test for time spent exploring novel object was less than 0.001 (between groups) with df = 4 and F (4, 35) = 8.588.

Referentially, the time spent exploring familiar object in young mice was 6.88 ± 3.04 s. One-year-old mice group was explored for 3.75 ± 0.97 s, while tomato group did for 3.83 ± 1.00 s and lemon group did for 4.29 ± 2.36 s. Mixture ingested group explored for 5.93 ± 2.63 s. The raw data showing amount of time exploring novel or familiar object were suggested as supplement data. 

### 3.3. Synergistic Improvement in Hippocampal Neurogenesis by Mixture of Tomato and Lemon Extracts in 1-Year-Old Mice

Tomato, lemon extract, or a mixture of both extracts were orally administered to 1-year-old female hairless mice for 6 weeks, and no changes were observed in body weight between groups (Figure 3). To investigate the effect of these extracts on hippocampal neurogenesis, we performed immunohistochemical staining for DCX, a marker of newly synthesized neurons.

As shown in Figure 4, the average count of DCX-positive cells (n = 3.63 ± 1.60) in 1-year-old mice was significantly reduced to an average of 8.7 ± 1.6%, compared to that (n = 41.43 ± 10.43, *p* < 0.001 compared with 1-year-old mice) in young mice. Tomato or lemon extract increased the number of DCX-positive cells by an average of 40.2 ± 2.6% (n = 5.08 ± 2.59, *p* = 0.91) or 49.4 ± 2.8% (n = 5.42 ± 2.85, *p* = 0.82), respectively, when compared to control 1-year-old mice. Interestingly, the mixture of tomato and lemon extracts increased the number of DCX-positive cells in the hippocampus by 94.3 ± 2.0% (n = 7.04 ± 1.99) compared to the 1-year-old control mice (*p* = 0.046), demonstrating the synergistic effect of the mixture on hippocampal neurogenesis. The *p*-value of ANOVA test was less than 0.001 (between groups) with df = 4 and F(4, 35) = 78.097. 

To quantify the newly synthesized neurons in the dentate gyrus, doublecortin (DCX) positive cells in hippocampal dentate gyrus were examined using immunohistochemistry. DCX-positive cells were counted in three randomly selected hippocampal sections per mice, and the administration of mixture extracts showed synergistic increment of DCX-positive cells compared to old mice. 

## 4. Discussion

In this study, we demonstrated that treating neuronal cells with a mixture of tomato and lemon extracts exhibited synergistic neuroprotective effects against oxidative stress-induced death. In addition, we observed that oral administration of a mixture of tomato and lemon extracts enhanced hippocampal neurogenesis significantly and synergistically in 1-year-old mice. We also demonstrated that the mixture of tomato and lemon extracts showed significant improvement in cognitive function in 1-year-old mice.

The cytoprotective effects of tomato and lemon have been previously demonstrated in several reports. Among the bioactive compounds in tomato, lycopene is a powerful antioxidant with oxygen quenching ability, which is two times higher than that of β-carotene and ten times higher than that of α-tocopherol [16]. Lycopene has the function of alleviating neurotoxicity, and long-term administration of lycopene is known to relieve mitochondrial oxidative damage, reduce neuro-inflammation, and improve BDNF expression along with enhanced memory function in amyloid beta-injected rodent model [17,18,19]. Citrus flavonoids are well-known for their antioxidant functions, especially via regulation of the Nrf2-antioxidase pathway [12]. 

Enhanced oxidative stress and damage associated with aging process eventually leads to cell death [20]. The brain is particularly vulnerable to such damage because it consumes high amount of oxygen with its limited antioxidant capacity. Several studies revealed that increased oxidative stress by aging process can cause neuronal dysfunction and mild cognitive impairment [21,22,23]. Therefore, treating antioxidants to scavenge free radicals has been suggested to prevent neuronal damage. 

The beneficial effects of tomatoes and lemons on hippocampal neurogenesis and cognitive function have been reported. Bae et al. reported that long-term administration of tomato extract increased hippocampal neurogenesis and cognitive function by stimulating the BDNF/ERK/CREB signaling pathway [12]. Hesperidin, the main flavonoid component of lemon, was shown to increase BDNF/TrkB signaling, thereby expressing the antidepressant effect in a mouse model [24]. Moreover, Nones et al. reported that hesperidin could increase the survival of neural progenitor cells and enhance neuronal population, suggesting its potential as a new therapeutic strategy for neurodegenerative diseases [25].

Our results indicate the synergistic cytoprotective effect of the mixture of tomato and lemon, which restored cell viability to the level of non-damaged cells. In addition, the stimulation of hippocampal neurogenesis was synergistically increased by ingestion of the mixture of tomato and lemon extracts. The mixture also significantly enhanced cognitive function estimated by NOR test results. It is interesting that the mixture of tomato and lemon showed significant change, while either tomato or lemon alone only showed a few increase of DCX-positive cells and slight elongation of duration to explore the novel objects. Although we cannot draw clear conclusions regarding the reasons for these outcomes, we assume that tomato and lemon supplement their compounds. As each of them contains a large amount of distinct nutrients, tomato can serve as a complement to lemon, and vice versa. Additionally, considering that pretreatment of the mixture effectively protected against oxidative stress, the preventive and proactive administration of tomato and lemon before the advanced degeneration may offer a more remarkable improvement in cognitive function.

The dose of “400 mg/kg” in mouse can be translated to 2 g/day in 60 kg weighted human. The amount of tomato and lemon used per daily dose in this study is extracted from and equivalent to 106 g of tomato and 60 g of lemon. The average weight of tomato ranges 180~200 g, while that of lemon ranges 110~120 g. If we develop the extracts into tablets as dietary supplement, 2 g per day, which is equal to 2 tablets twice daily, is suitable for compliance because a tablet the size of one fingertip usually weighs 500 mg. 

The results of our study should be interpreted in acknowledgement of some limitations. First, the sample size was small, thus limiting the generalizability of the results. Additionally, possible confounders that can affect object exploration, such as anxiety of the mouse, should be considered. In addition, we did not measure oxidative stress parameters in in vivo models. Future studies, that include measurement of total distance travelled by mouse to estimate the anxiety level of mouse and checking antioxidant effect of the extracts, would help to confirm the role of tomato and lemon extracts in improving cognitive function. In conclusion, the mixture of tomato and lemon extract attenuated cellular oxidative stress, stimulated hippocampal neurogenesis and enhanced cognitive function in mice. As our study is based on cellular and murine-based experiments, future clinical studies to confirm the improved effects of tomato and lemon extract mixture on human cognitive decline are needed. Additionally, further studies would reveal the active ingredients for neurogenesis. 

## 5. Conclusions

This study demonstrates that tomato and lemon extracts, with synergistic effect, can reduce cellular oxidative stress and increase novel object recognition probably due to enhanced neurogenesis. 

## Figures and Tables

**Figure 1 foods-11-03418-f001:**
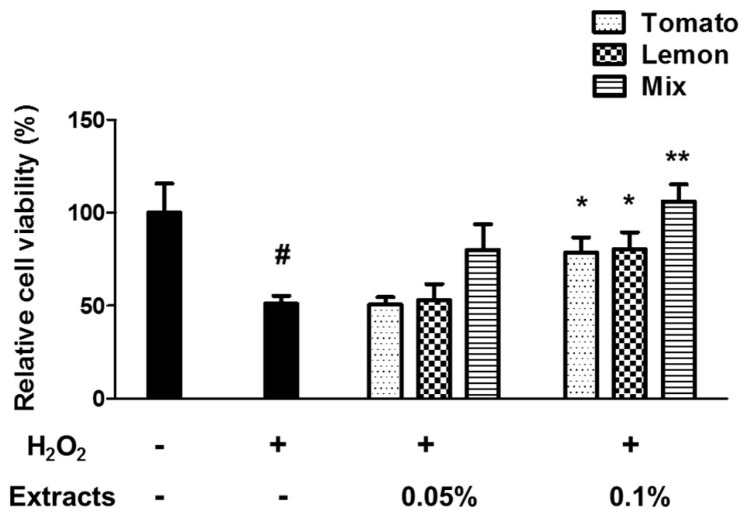
Protective effect of single extract with tomato or lemon and their mixture on cell viability. PC12 cells were pre-treated with tomato, lemon, or mixture extract at 0.05% and 0.1% concentrations for 24 hours, and were exposed to 100 μmol/L H_2_O_2_ for 30 min. Cell viability was measured using WST-1 assay. Relative cell viability compared to control (100%) was shown as percentage. ^#^
*p* < 0.05 vs. untreated control group, * *p* < 0.05, ** *p* < 0.01 vs. H_2_O_2_-treated group. T: tomato, L: lemon, T + L: mixture of tomato and lemon.

**Figure 2 foods-11-03418-f002:**
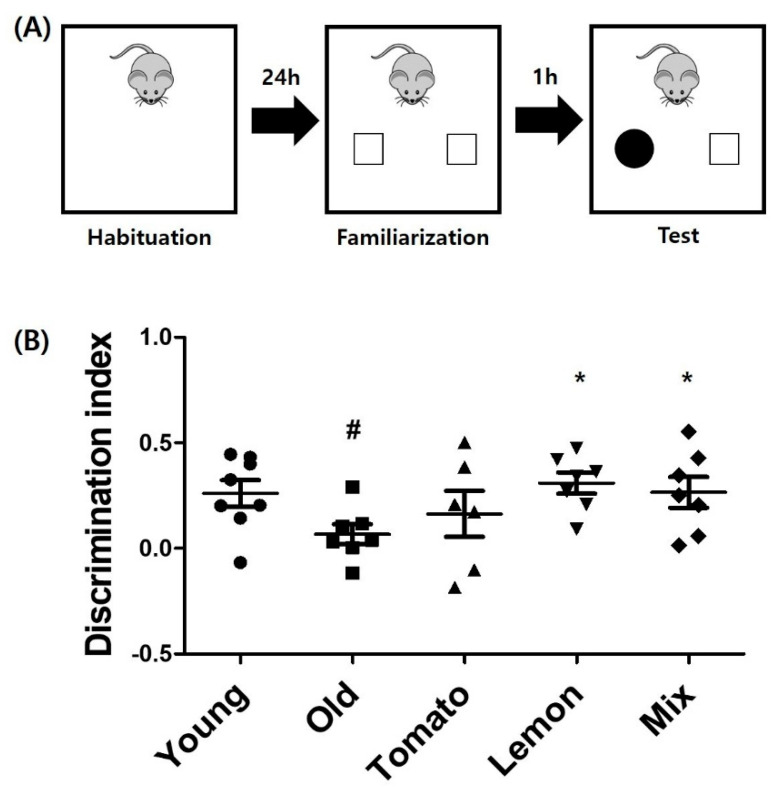
Enhanced novel object recognition by tomato, lemon or mixture extracts. (**A**) Novel objective recognition test. Mice were allowed to explore the open field box for habituation, and two identical objects were placed in the box for familiarization. One hour after familiarization, one object was replaced with new object and novel object recognition test was performed for 5 min. (**B**) Time spent to explore novel object was measured for each group. One-year-old mice showed significantly shorter duration of curiosity than young mice. The mixture ingested group showed significantly increased interest for new object, compared to old control group. ^#^
*p* < 0.01 vs. young mice, * *p* < 0.05 vs. old control group. n = 8 for each groups. Discrimination index (DI) was depicted at the bottom. DI allows discrimination between the novel and familiar objects and is calculated as the difference in exploration time for familiar object divided by the total amount of exploration of the novel and familiar objects [DI = (T_novel_ − T_familiar_)/T_novel_ + T_familiar_].

**Figure 3 foods-11-03418-f003:**
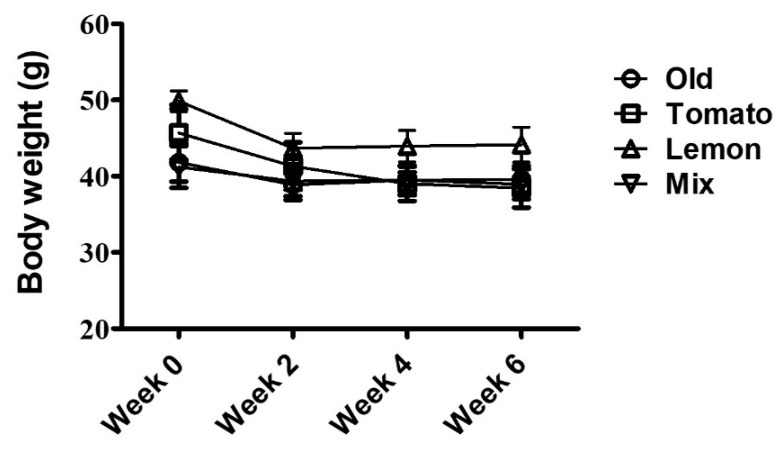
There was No change in body weight during the study period.

**Figure 4 foods-11-03418-f004:**
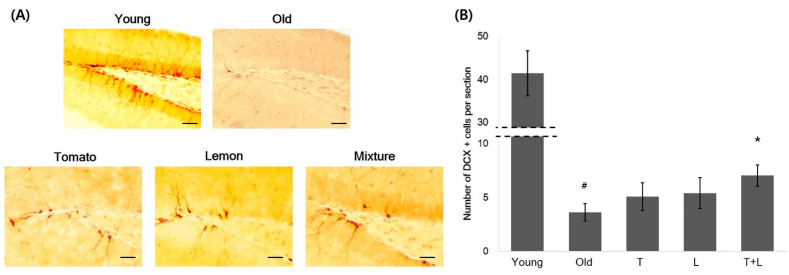
Increase of DCX-positive hippocampal neurons by oral administration of tomato, lemon and mixture extracts. (**A**) Representative images of DCX-positive neurons. Scale bars: young, old, tomato, lemon, mixture, 100 μm. (**B**) The average DCX-positive cells per section was calculated and represented as a bar graph (n = 8 mice per group). The administration of mixture extracts showed synergistic increment of DCX-positive cells compared to old mice. ^#^
*p* < 0.01 vs. young mice. * *p* < 0.05 vs. old mice. T: tomato, L: lemon, T + L: mixture of tomato and lemon.

## Data Availability

The data used to support the findings of this study are available from the corresponding author upon request.

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
