# Peer review of "Mixture of Tomato and Lemon Extracts Synergistically Prevents PC12 Cell Death from Oxidative Stress and Improves Hippocampal Neurogenesis in Aged Mice"

_foods, 2022, doi:10.3390/foods11213418_

Round 1
Reviewer 1 Report
1- Abstract: It lacks the number of mice used in the study ----
2- Introduction:
- language should be revised to be more formal. “Meanwhile, there is a proverbial saying: You are what you eat. This notion highlights the close relationship between dietary habits and health. Typically, a high-fat and high- salt diet can cause cerebrovascular diseases and metabolic syndrome -------"
- Abbreviations should be mentioned in full name for first time “BDNF”
- Please revise and rephrase this sentence “To investigate the synergistic effects of tomato and lemon extract in aging-associated cognitive decline, we orally administered tomato extract, lemon extract, and a mixture of these two extracts to 1-year-old mice for 6 weeks, and analyzed hippocampal neurogenesis and cognitive function.”
3- Materials and methods
- In the cell viability assay : the authors should mention the concentrations of tomato, lemon and their mixture allowed for each well.
- Line 105 “we had experience with in the previous study of neurogenesis” please revise this sentence grammatically.
- The average weight of the mice used in the experiment should be mentioned.
- The authors should mention reference for the basis of dose conversion from human to mice.
- The authors should give information about the number of slides and fields examined to analyze IHC of DCX.
-
4- Results
- Line 224 were examined
- IHC figure should be colored
5- Discussion:
- The discussion is speculative
I think the manuscript lacks the probable analysis of the 2 extracts active ingredients that could be linked to their neurogenesis potential. Also oxidative stress parameters are lacking. May be
DPPH Assay is beneficial in current study.
6- Conclusion
It depends on oxidative stress elimination speculation and nothing in in vivo experiment handled the antioxidant effect of tomato or lemon extract.
Author Response
Thank you so much for your precious and impressive comments and opinion about this article. All the changes and responses to reviewers’ comments were designated by underlines with highlighted color in main text.
We, the authors, are truly appreciating all the guidance and encouragement of yours. As for the things you indicated, I did my best in correcting all of them. But if you still see any problem or error, please just let me know and give your comments. I always appreciate your review. Thank you again.
---
1- Abstract: It lacks the number of mice used in the study ----
- We mentioned the number of mice used in the study to the abstract.
2- Introduction:
- language should be revised to be more formal. “Meanwhile, there is a proverbial saying: You are what you eat. This notion highlights the close relationship between dietary habits and health. Typically, a high-fat and high- salt diet can cause cerebrovascular diseases and metabolic syndrome -------"
è We revised the sentence to be more formal. Thank you.
- Abbreviations should be mentioned in full name for first time “BDNF”
è We described the full name ahead of its abbreviation. Thank you.
- Please revise and rephrase this sentence “To investigate the synergistic effects of tomato and lemon extract in aging-associated cognitive decline, we orally administered tomato extract, lemon extract, and a mixture of these two extracts to 1-year-old mice for 6 weeks, and analyzed hippocampal neurogenesis and cognitive function.”
è We revised and rephrased the expression. Thank you.
3- Materials and methods
- In the cell viability assay : the authors should mention the concentrations of tomato, lemon and their mixture allowed for each well.
è We additionally mentioned the concentrations used in each well.
- Line 105 “we had experience with in the previous study of neurogenesis” please revise this sentence grammatically.
è We omitted the expression because it was not necessary.
- The average weight of the mice used in the experiment should be mentioned.
è We added figure 3 to show that there was no change to the average body weight of the mice and no difference among groups. We also mentioned the average weight of the mice.
- The authors should mention reference for the basis of dose conversion from human to mice.
è We added the reference of dose conversion from human to mice.
- The authors should give information about the number of slides and fields examined to analyze IHC of DCX.
è To quantify the total number of DCX-positive cells in the hippocampal dentate gyrus, cell counts were performed in at least three consecutive sections at a magnification of 40x us-ing a counting grid, defining an area of interest to a width of 500μm along the fissure by a blinded examiner.
4- Results
- Line 224 were examined
è We changed ‘was’ to ‘were’. Thank you.
- IHC figure should be colored
è We changed to the colored figures.
5- Discussion:
- The discussion is speculative
I think the manuscript lacks the probable analysis of the 2 extracts active ingredients that could be linked to their neurogenesis potential. Also oxidative stress parameters are lacking. May be
DPPH Assay is beneficial in current study.
- Thank you for the comments. Now that you mentioned it, measuring oxidative stress parameters would be worthy finding the possible mechanism of neurogenesis and cellular protection. We also agree with you that the analysis with active ingredients is lacking, but as a pilot study, we could not specify the active ingredients yet. We are planning to find which ingredients are key to the neurogenesis in our further study, and we would apply DPPH assay as you recommended then. We pointed out these in the limitation.
6- Conclusion
It depends on oxidative stress elimination speculation and nothing in in vivo experiment handled the antioxidant effect of tomato or lemon extract.
- Thank you for the precious comments. We would further suggest oxidative stress parameters in our future study.

Reviewer 2 Report
Comments and suggestions for authors
The manuscript entitled “Mixture of tomato and lemon extracts synergistically prevents PC12 cell death from oxidative stress and improves hippocampal neurogenesis in aged mice“ which is presented by Ji Yeon Hong and collaborators, has interesting results. Overall, the results are solid and exhibit the strong relationship between the mixture of tomato and lemon extracts and its neuroprotective in aged mice. The manuscript is well written. However, I have a few concerns that need to be addressed in the revised version.
Major comment
1. Regarding to the active compounds in the tomato and lemon extract, in the discussion part, the author mentioned about the active and major compounds that found in the tomato and lemon extract (in the references). It would be better if the author can provide the quantification of the active compounds that found in each extract and their mixture. It is important for discussion about the pharmacological activities and their active compounds and for the quality control of this mixture formula.
2. The authors mentioned that the tomato and lemon mixture extract improved the cognitive impairment in old mice and the underlying mechanism were involved in the increasing of the DCX positive cell in dentate gyrus of hippocampus. Only single mechanism has been done. The authors need to clarify the other related mechanism that can be explained for the pharmacological activity of this mixture extracts.
Minor comment
1. The total number of mice should be reported in the method and the number of mice in each group should be reported too.
2. Figure 1, the significant differences that showed in the graph should be expressed in the proper way as shown in figure 2 or figure 3.
3. The behavioral results should be reported in 3.2 and Figure 2. Immunohistochemistry results should be reported in 3.3 and Figure 3.
4. In immunohistochemistry results, the author should place the scale ruler (such as 100 uM) in the figure.
1. The author should add the author name of the plant species eg. Lycopersicon esculentum Mill., Citrus limon (L.) Burn f.
Author Response
Thank you so much for your precious and impressive comments and opinion about this article. All the changes and responses to reviewers’ comments were designated by underlines with highlighted color in main text.
We, the authors, are truly appreciating all the guidance and encouragement of yours. As for the things you indicated, I did my best in correcting all of them. But if you still see any problem or error, please just let me know and give your comments. I always appreciate your review. Thank you again.
----
The manuscript entitled “Mixture of tomato and lemon extracts synergistically prevents PC12 cell death from oxidative stress and improves hippocampal neurogenesis in aged mice“ which is presented by Ji Yeon Hong and collaborators, has interesting results. Overall, the results are solid and exhibit the strong relationship between the mixture of tomato and lemon extracts and its neuroprotective in aged mice. The manuscript is well written. However, I have a few concerns that need to be addressed in the revised version.
Major comment
- Regarding to the active compounds in the tomato and lemon extract, in the discussion part, the author mentioned about the active and major compounds that found in the tomato and lemon extract (in the references). It would be better if the author can provide the quantification of the active compounds that found in each extract and their mixture. It is important for discussion about the pharmacological activities and their active compounds and for the quality control of this mixture formula.
à Thank you for the precious comments. The active and major compounds mentioned in discussion were not quantified in this pilot study. We would like to specify the active ingredient effective for neurogenesis in our future experiment.
- The authors mentioned that the tomato and lemon mixture extract improved the cognitive impairment in old mice and the underlying mechanism were involved in the increasing of the DCX positive cell in dentate gyrus of hippocampus. Only single mechanism has been done. The authors need to clarify the other related mechanism that can be explained for the pharmacological activity of this mixture extracts.
à Thank you and we agree with your idea. We think the increment of the neuronal cells is strongly related to the improved cognitive function, although it is a single mechanism. We also assume that antioxidant effects would enhance the impaired cognitive function, which is not handled in the mouse model. We added this point to the discussion that measuring oxidative stress parameters in the murine model would help explaining the related mechanism.
Minor comment
- The total number of mice should be reported in the method and the number of mice in each group should be reported too.
à A total of 40 mice, 8 in each group, was used in the experiment.
- Figure 1, the significant differences that showed in the graph should be expressed in the proper way as shown in figure 2 or figure 3.
à Figure 1 format was edited. Thank you.
- The behavioral results should be reported in 3.2 and Figure 2. Immunohistochemistry results should be reported in 3.3 and Figure 3.
à We changed the order as you recommended.
- In immunohistochemistry results, the author should place the scale ruler (such as 100 uM) in the figure.
à Scale bar was inserted in the figure.
- The author should add the author name of the plant species eg. Lycopersicon esculentumMill., Citrus limon (L.) Burn f.
à Thank you for the comments. We added the author name.

Reviewer 3 Report
This study is a descriptive study. The study shown that extracts of tomato and lemon have a dose dependent neuroprotective effect when used alone, no effect at the concentration of 0,05% but effect at 0.1%. So, the authors should set forth the purpose of mixture of tomato and lemon extracts.
Neither tomato nor lemon extract increased the number of DCX-positive cells compared to control 1-year-old mice. Is it related to insufficient dose?
Author Response
Thank you so much for your precious and impressive comments and opinion about this article. All the changes and responses to reviewers’ comments were designated by underlines with highlighted color in main text.
We, the authors, are truly appreciating all the guidance and encouragement of yours. As for the things you indicated, I did my best in correcting all of them. But if you still see any problem or error, please just let me know and give your comments. I always appreciate your review. Thank you again.
---
Reviewer 3
This study is a descriptive study. The study shown that extracts of tomato and lemon have a dose dependent neuroprotective effect when used alone, no effect at the concentration of 0,05% but effect at 0.1%. So, the authors should set forth the purpose of mixture of tomato and lemon extracts.
- As we mentioned in methods section, the purpose of using the mixture of tomato and lemon extracts was to develop as dietary supplement. We previously screened 133 items of food materials (commonly consumed fruits, grains, and vegetables) as possible candidates and finally selected tomato and lemon which increased BDNF, FGF-1 and IGF-1. These hormones are known to induce neurogenesis.
Neither tomato nor lemon extract increased the number of DCX-positive cells compared to control 1-year-old mice. Is it related to insufficient dose?
- Although it was not statistically significant, tomato or lemon extract increased the number of DCX-positive cells compared to 1-year-old mice. We think this was perhaps due to small number of mice. If it was related to insufficient dose, the number of DCX-positive cell would be similar to control group.

Round 2
Reviewer 1 Report
Abstract:
- Line 29 and 30 “novel object recognition” replace NOR
Introduction:
- Language revision is required; it is informal to mention our
Material and method:
- Line 116 start with Twelve instead of 12
-
Author Response
Abstract:
- Line 29 and 30 “novel object recognition” replace NOR
--> We replaced with NOR.
Introduction:
- Language revision is required; it is informal to mention our
--> We changed from 'our' to 'the'.
Material and method:
- Line 116 start with Twelve instead of 12
--> We change to 'Twelve'.
Thank you for the comments.
Reviewer 2 Report
The authors have modified the manuscript following reviewer’s comments. I accept this manuscript for publication.
Author Response
Thank you for the acceptance.
Reviewer 3 Report
accept
Author Response
Thank you for the acceptance.